# Development and Validation of Reverse Transcriptase Loop-Mediated Isothermal Amplification (RT-LAMP) as a Simple and Rapid Diagnostic Tool for SARS-CoV-2 Detection

**DOI:** 10.3390/diagnostics12092232

**Published:** 2022-09-15

**Authors:** Ahmad M. Aldossary, Essam A. Tawfik, Musaad A. Altammami, Azzam A. Alquait, Rayan Y. Booq, Bandar K. Sendy, Mohammed S. Alarawi, Takashi Gojobori, Asmaa M. Altamimi, Taghreed A. Alaifan, Ahmed M. Albarrag, Essam J. Alyamani

**Affiliations:** 1National Center of Biotechnology, King Abdulaziz City for Science and Technology (KACST), Riyadh 12354, Saudi Arabia; 2Computational Bioscience Research Center, King Abdullah University of Science and Technology (KAUST), Thuwal 23955, Saudi Arabia; 3Public Health Laboratory, Public Health Authority, Riyadh 13354, Saudi Arabia

**Keywords:** Reverse Transcriptase Loop-Mediated Isothermal Amplification (RT-LAMP), colorimetric assay, COVID-19, SARS-CoV-2, *ORF1a* gene, *N* gene

## Abstract

Since the COVID-19 pandemic outbreak in the world, many countries have searched for quick diagnostic tools to detect the virus. There are many ways to design diagnostic assays; however, each may have its limitations. A quick, sensitive, specific, and simple approach is essential for highly rapidly transmitted infections, such as SARS-CoV-2. This study aimed to develop a rapid and cost-effective diagnostic tool using a one-step Reverse Transcriptase Loop-Mediated Isothermal Amplification (RT-LAMP) approach. The results were observed using the naked eye within 30–60 min using turbidity or colorimetric analysis. The sensitivity, specificity, and lowest limit of detection (LoD) for SARS-CoV-2 RNA against the RT-LAMP assay were assessed. This assay was also verified and validated against commercial quantitative RT-PCR used by health authorities in Saudi Arabia. Furthermore, a quick and direct sampling from the saliva, or buccal cavity, was applied after simple modification, using proteinase K and heating at 98 °C for 5 min to avoid routine RNA extraction. This rapid single-tube diagnostic tool detected COVID-19 with an accuracy rate of 95% for both genes (*ORF1a* and *N*) and an LoD for the *ORF1a* and *N* genes as 39 and 25 copies/reaction, respectively. It can be potentially used as a high-throughput national screening for different respiratory-based infections within the Middle East region, such as the MERS virus or major zoonotic pathogens such as *Mycobacterium paratuberculosis* and *Brucella* spp., particularly in remote and rural areas where lab equipment is limited.

## 1. Introduction

In late 2019, Wuhan in Hubei Province in China reported an unknown fast-spreading viral outbreak as the world’s fastest pandemic disease in the twenty-first century. The World Health Organization (WHO) declared it soon after as COVID-19 [1]. It has become a global public healthcare, social and economic burden. Symptoms of this infection include but are not limited to high fever, dry cough, fatigue, myalgia, severe pneumonia, acute respiratory distress, and death [2]. In the past, coronaviruses (CoVs) have caused two serious outbreaks: Severe Acute Respiratory Syndrome (SARS-CoV) and Middle East Respiratory Syndrome (MERS) [3]. In 2002, a subtype of the beta-COV rapidly spread across Guangdong, China. This outbreak resulted in 8000 infections and 774 fatalities in 37 countries [4]. In 2012, MERS-CoV was first detected in Saudi Arabia, killing around 858 people, with over 2494 confirmed cases [5]. 

Despite the efforts to develop antiviral drugs for COVID-19 and the global coverage of vaccination, the incidence rate is high in many countries, including the United States, where more than 85,007,630 infected cases have been detected, with 1,002,946 deaths. By 17 June 2022, SARS-CoV-2 infection had rapidly spread and had infected more than 535,863,950 people, with more than 6,314,972 deaths reported worldwide. In Saudi Arabia, there have been 781,168 infected cases and a total of 9179 deaths up to that date (i.e., June 17) [6]. Owing to the continuing and fast spreading of the viral infection, efforts to continue to develop quick, sensitive, specific, cost-effective, and rapid diagnostic tools are still urgently needed to facilitate the early detection of positive cases in the hope of controlling its spreading, particularly in poor and rural parts of the world. The WHO and CDC (i.e., Centers for Disease Control and Prevention) recommend using RT-PCR technology and serology testing to identify positive COVID-19 cases [7,8]. Many companies have adopted and developed these approaches and have introduced many RT-PCR and antibody-based kits to the market and clinical settings. However, those technologies have many limitations that could make them inaccessible to many parts of the world. Such limitations can include the need for high technology, time-consuming, special reagents that may not be available constantly, specialized laboratories, trained technicians, and in some cases, low accurate and false-positive results [9,10].

Considering the previously mentioned drawbacks, sample processing and management challenges for the frontline workers against COVID-19, the continuing fast spreading of infection, the time taken for each test, and the cost of testing, the development of a simple yet specific and rapid diagnostic tool is still desired. Thus, this study underpins Reverse Transcriptase Loop-Mediated Isothermal Amplification (RT-LAMP) for easy and fast diagnosis of COVID-19. LAMP is a quick, specific, sensitive, and cost-effective diagnostic technology that could replace current RT-PCR. This technology only needs a heat block or water bath to maintain a temperature of 65 °C to conduct a reaction. A DNA polymerase with chain displacement activity (Bst) is coupled with reverse transcription enzyme activity in one single tube that can be developed and optimized for rapid detection of COVID-19. The reaction results can be visualized after 30–60 min by observing either the turbidity or colorimetric analysis using the naked eyes or under the UV lamp [11,12,13].

Several studies have reported the successful use of LAMP for the rapid and specific detection of COVID-19. Lu et al. developed this simple and rapid assay for SARS-CoV-2 detection using the viral *N* gene [14]. The results showed that this technique has a high sensitivity, specificity, and reproducibility, with a limit of detection (LoD) equal to 118.6 copies of SARS-CoV-2 RNA per 25 μL reaction. The reaction can also be monitored and detected visually after 40 min when the color changes from red to yellow. Finally, the viability of this assay was evaluated by comparing the RT-LAMP and RT-qPCR assays using 56 clinical samples. The RT-LAMP assay perfectly agreed with the RT-qPCR assay for the tested samples [14]. Another RT-LAMP was developed by Chow et al. using SARS-CoV-2 isolate and respiratory samples from patients with COVID-19 or other respiratory viral infections [B]. The LoD for this LAMP assay is 42 copies/reaction, with a sensitivity of 95.07% and 98.21% for 60 and 90 min of incubation, respectively. The study also tested the sensitivity of different respiratory sampling. It was found that the nasopharyngeal swabs had >96% compared to sputum (or deep throat saliva) >94% and throat swab >93% samples. The results of the other respiratory viruses showed negative for all tested samples (n = 143), indicating 100% specificity [15]. Roumani et al. have also developed and evaluated RT-LAMP for COVID-19 detection compared to RT-qPCR for 152 clinical nasopharyngeal swabs. The finding indicated that both techniques have a good concordance, with the RT-LAMP having a high specificity (99%) and lower sensitivity (63.3%) compared to the RT-qPCR [16]. Anastasiou et al. tested using LAMP to detect SARS-CoV-2 genomes directly in respiratory samples without further extracting the viral nucleic acids beforehand. The results indicated that this assay has a 100% specificity and a moderate sensitivity of 68.8% for the direct testing of nasopharyngeal swabs in a viral transport medium [17]. These studies showed the potential use of the inexpensive, highly specific, and moderate sensitive RT-LAMP for SARS-CoV-2 detection owing to its rapid deployment and mobility, especially in resource-limiting areas.

Furthermore, Diego and colleagues have successfully developed a rapid detection of SARS-CoV-2 targeting both *ORF1a/b* and *N* genes. The findings indicated that this assay is affordable (≈1.7 €/sample), with a rapid detection of under 25 min [18]. A further study by the same Diego et al. showed that the developed RT-LAMP (i.e., N15-RT-LAMP) could detect SARS-CoV-2 RNA from the naso-oropharyngeal swabs but also from the urine. This technique was used to test the urine of 300 patients divided into three groups: RT-qPCR COVID-19 positive using a nasopharyngeal swab (n = 100), COVID-19 negative (n = 100), and COVID-19 positive with disease recovery (n = 100). The results showed a trace amount of SARS-CoV-2 RNA, with only 4% positivity found in the positive group [19]. Other studies continued to prove the reliability of LAMP assay in detecting the SARS-CoV-2 virus using different sampling locations. Saliva has emerged as a practical alternative to nasopharyngeal swab samples. Kobayashi et al. developed a direct RT-LAMP from saliva samples of symptomatic patients. The findings indicated an overall sensitivity of 77.2%. For samples with >102 copies/μL, the sensitivity was enhanced to 93.2%, and the specificity was reported as 97%. Further tests in the saliva samples demonstrated that the viral load peaked in the first days of symptoms and reduced afterward [20]. Another study by Rajh and colleagues evaluated an extraction-free RT-LAMP for detecting SARS-CoV-2 in saliva samples compared to RT-qPCR. The sensitivity and specificity of the saliva-based RT-qPCR were >95% and 99%, respectively, while the sensitivity and specificity of the saliva-based RT-LAMP were >70% and >93%, respectively [21]. Finally, Londono-Avendano et al. evaluated the detection of SARS-CoV-2 in nasopharyngeal swabs, aspirates, and saliva using the LAMP assay. The colorimetric detection was attained either by the naked eye or by an image analysis mobile application. The overall detection accuracy was enhanced when using the mobile application (75–86%) compared to the naked eye (61–74%). In contrast, the detection accuracy was 55, 70, and 80% for the nasopharyngeal swabs, aspirates, and saliva, respectively [22]. The abovementioned studies illustrated the possibility of using RT-LAMP with different sampling locations than the naso-oropharyngeal swabs. It is important to further optimize the direct application of samples to diagnostic assays, thus allowing high-throughput, rapid, and easily implemented patient screening for COVID-19.

Several studies have reported the successful use of LAMP for detecting SARS-CoV-2, yet it is still not widely used clinically in many countries. A study performed in Argentina evaluating several commercially available RT-qPCR and RT-LAMP assays against SARS-CoV-2 *E, N,* and *RdRp* genes was reported by Fellner et al. [23]. The RT-qPCR assays demonstrated 100% specificity and high sensitivity, with the *E* and *N* genes providing greater sensitivity, while all three genes can improve the specificity. The RT-LAMP assays showed specificity and sensitivity for the *N + ORF1* genes as 100% and >75%, respectively, while the specificity and sensitivity were 83.9% and 100% for the *E* gene, respectively. This variability of the diagnostic assay performance suggested that selecting the correct gene can be a crucial step when developing the most appropriate diagnostic test [23]. In Saudi Arabia, El-Kafrawy et al. tested the RT-LAMP kit for detecting SARS-CoV-2 utilizing a combination of three target genes (*ORF1ab*, *N*, and *S*). The results demonstrated that the RT-LAMP has a 100% concordance in both sensitivity and specificity with RT-PCR for detecting SARS-CoV-2 nucleic acid, with a reaction time of 30 min for the RT-LAMP assay [24]. Therefore, in this study, a colorimetric single-tube RT-LAMP assay was furtherly optimized and validated as a screening method to detect SARS-CoV-2 viral RNA using two targeted genes (*ORF1ab* and *N* genes) in the Saudi population. In addition, a quick and direct sampling from the saliva, or buccal cavity, was applied after simple modification to avoid the routine and time-consuming RNA extraction method.

## 2. Materials and Methods

### 2.1. Target Selection and RT-LAMP Primers Design

The total size of the SARS-CoV-2 genome is 30 kb in GenBank (Accession number: MN908947). The genome of the virus contains ten open reading frames (ORFs). The first and largest one is *ORF1a/b*, which encodes for two big polyproteins with a size of 1888 bp. Another vital gene considered in this study and presented in the SARS-CoV-2 genome is the *N* gene, which encodes for structural protein “nucleocapsid phosphoprotein” with 1260 bp in size. These two targets have been recommended by the Chinese CDC and published lately [25]. Therefore, these two targets were used due to their high level of specificity in designing the RT-LAMP test. The two genes (*ORF1a* and *N* gene) were synthesized in vitro by DNA Technologies, Inc. (IDT), Coralville, IA, USA. Positive controls were synthesized for the nucleocapsid gene (*N* gene, 1260 bp) and partial *ORF1* (510 bp). The LAMP technology uses six specific primers targeting eight different sequences on the gene of interest. Six specific LAMP primers were designed to target a small fragment of *ORF1a, N* genes and Ribonuclease P (RNAse P), as demonstrated in the Appendix A, respectively, using an online primer Explorer V5 (Eiken Chemical Co., Ltd., Tokyo, Japan; https://primerexplorer.jp/e/ (accessed on 7 July 2020)).

### 2.2. Synthesis of SARS-CoV-2 RNA

To synthesize RNA from the control DNA fragments (*ORF-1a* and *N*), primers were designed to amplify 510 bp and 722 bp from *ORF1a* and *N*, respectively. T7 promoter sequence has been added to the forward primer (5′-TAATACGACTCACTATAGGGAGA-3′) to facilitate the RNA transcription. The PCR was performed in two-cycle conditions using SeqAmp DNA Polymerase (Takara Bio, San Jose, CA, USA). In the first 15-cycle, the target region is amplified, followed by 20-cycle to incorporate T7 in the amplicon. The forward primer for *ORF-1a* (5′-TAATACGA CTCACTATAGGGAGA-TTCAACCAAGGGTTGAAAAG-3′) and the reverse (5′-GTATTTCAAGAAGGTTGTCATT AAG-3′) while the forward primer for *N* (5′-TAATACGACTCACTATAGGGAGA-TGGACCCCAAAATCAGCG-3′) and the reverse (5′-TTTGGCCTTGTTGTTGTTG-3′) were used to amplify the target. The amplicons were purified using QIAquick PCR Purification Kit (Qiagen, Germantown, MD, USA), followed by RNA transcription using a HiScribe™ T7 High Yield RNA Synthesis Kit (New England Biolabs, Hitchin, UK), per the manufacturer protocol. The transcribed RNA was treated with 1 μL DNase I reagent (incubate at 37 °C for 10 min), followed by the purified step using Monarch RNA Cleanup Kit (New England Biolabs, Hitchin, UK), and then quantified by the SpectraMax QuickDrop spectrophotometer (Molecular Devices, San Jose, CA, USA).

### 2.3. Development of the RT-LAMP Assay

The colorimetric RT-LAMP assay was performed in 25 μL reaction volume using WarmStart^®^ colorimetric LAMP 2X Master Mix (DNA and RNA) (New England Biolabs, Hitchin, UK). The reaction was designed to provide a one-step LAMP of RNA (RT-LAMP) targets. The kit contains a blend of *Bst* 2.0 WarmStart DNA Polymerase and WarmStart RTx Reverse Transcriptase in an optimized LAMP buffer solution. The reaction includes 2.5 μL of 10× primer mix (16 μM each of FIB and BIP, 2 μM each of F3 and B3, 4 μM each of LF and LB primers), 12.5 μL of WarmStart LAMP 2X master mix, 1 μL of the sample and the reaction was brought up to 25 μL by doubled distilled water (Millipore, Bedford, MA, USA). The reaction was then vortexed and collected again by quick centrifugation before it was sealed and placed in a heat block at 65 °C for 30–60 min. The reaction moved to PCR Cooler Rack (or ice) for 5–10 s and then observed visually against the contained pH indicator. The pink color signifies a negative reaction, while the yellow color confirms a positive reaction.

### 2.4. Validation of the RT-LAMP Assay

Different RNA samples were used to validate and test the developed RT-LAMP assay, including clinically isolated RNA samples. The validation was carried out using blind clinical samples, i.e., nasopharyngeal swabs, to test if the new assay concordant with the RT-PCR commercial kit results. This validation study was performed by the Saudi CDC (i.e., Public Health Authority), Riyadh, Saudi Arabia, after obtaining an ethical approval number of H1RI-15-Nov20-01 from the King Saud Medical City (KSMC) institutional review board (IRB), Riyadh, Saudi Arabia, for using patients’ biological samples. Different clinical samples (3 μL each) were run using RT-PCR and RT-LAMP side by side to confirm the validation of the latter tool. The level of sensitivity, specificity, and reproducibility was estimated accordingly. The LoD in RNA copy number, positive predictive value (PPV), and negative predictive value (NPV) were computed to validate the developed RT-LAMP assay.

### 2.5. Optimizing the RNA Extraction from Clinical Samples to the RT-LAMP Assay (Point of Care Testing)

The RNA extraction from saliva and the buccal swab was completed by transferring 500 μL of phosphate normal saline (PBS) pH 7.4 (Cat. No.10010031, ThermoFisher Scientific, Waltham, MA, USA) to the samples before RNA extraction. A 50 μL sample was added to an equivalent volume of PBS (50 μL) alone or with 5 μL proteinase K (Qiagen, Hilden, Germany). In addition, 1M Tris-HCl buffer with a pH 8.0 (Cat. No.15568025, ThermoFisher Scientific Waltham, MA, USA) was used as an alternative buffer to PBS. Furthermore, RNA from the samples extracted by Monarch Total RNA. Miniprep Kit (New England Biolabs, Hitchin, UK) according to the manufacturer’s protocol.

## 3. Results and Discussion

### 3.1. RT-LAMP Assay Design and Testing against DNA Control Samples

This study aimed to develop a one-step RT-LAMP assay to detect SARS-CoV-2 RNA. This assay has numerous advantages compared to other available detection assays, including its simplicity, rapidity, and cost-effectiveness, with no direct effect on the sensitivity or specificity of the results. The developed RT-LAMP assay in this study was designed to target two genes, *ORF1a* and *N*, in the SARS-CoV-2 virus (Figure 1). Five target sets were designed to explore the best target in each gene (A, B, C, D, and E) with LAMP primers (F3, B3, FIP, BIP, LF, and LB). All targets are first tested with a synthesized DNA fragment to ensure the assay works. With an incubation period of 30 min at 65 °C, all targets in the *ORF1a* and *N* genes gave positive results (yellow color) with 9 × 10^9^ and 2 × 10^9^ DNA copies, respectively. The color for the control sample, i.e., with no DNA template, remained pink (Figure 2).

A serial dilution of the DNA fragment was performed to assess the LoD for the DNA fragment with the RT-LAMP assay. The sensitivity of *ORF1a* was assessed from 9 × 10^8^ to 0.09 copies/reaction, while the evaluation for the LAMP assay for the *N* gene started from 200,000 to 2 copies/reaction. Figure 3 and Figure 4 show that the primers targeting the B region in the *ORF1a* and *N* genes are highly sensitive as they can detect 1 copy and 200 copies of the DNA fragment, respectively. The change in RT-LAMP assay color is also confirmed by gel electrophoresis (Figure 3 and Figure 4).

The primer design is a crucial step for nucleic acid amplification by PCR. The primer design for the LAMP assay is more complicated than the PCR since 6 primers are needed. Not all targets in the *ORF1a* and *N* genes resulted in the same detection sensitivity, which could be related to the nature of the target gene or the primer’s design. Five targets were initially identified in the *ORF1a* and *N* genes (A, B, C, D, and E). The five targets related to the *N* gene were evaluated with similar primers to the previous studies [26,27,28,29]. In addition, the *ORF1a* was explored as a potential target with RT-LAMP assay [2,30,31]. However, new targeted regions in the *ORF1a* gene were assessed in this study, which was not reported previously. This assay evaluated the primer’s specificity in the B region in silico against SARS-CoV-2 or SARS-CoV-1 on the same targeted genes. The B target on the *ORF1a* showed more specificity for SARS-CoV-2 in terms of mismatching with a similar coronavirus (SARS-CoV-1) with 54 mismatches from 204 bases. On the other hand, the B target on the *N* gene exhibited a more similarity to SARS-CoV-1, with 20 mismatches from 213 bases showing less specific primer binding (Figure 5). Further assay specificity is required for the B target in the *N* gene.

### 3.2. Testing the RT-LAMP Assay against the Synthesized RNA Samples

After evaluating the developed RT-LAMP assay against synthesized DNA fragments, testing against synthesized RNA samples was performed by adding a T7 promoter to the DNA PCR fragment, followed by in vitro RNA synthesis (Figure 6). The size of the RNA for the *ORF1a* gene was 510 bases, while it was 722 bases for the *N* gene. The RT-LAMP assay was evaluated to detect 3.9 × 10^6^ RNA fragment copies per reaction, going down to 3.9 copies for the *ORF1a* gene (Figure 7A), and 2.5 × 10^6^ RNA fragment copies per reaction, going down to 2.5 copies for the *N* gene (Figure 7B). The results in Figure 7 demonstrated that primers targeting the A and B regions in the *ORF1a* gene, and A-D regions in the *N* gene, are highly sensitive as they can detect 39 copies and 25 copies of the RNA fragment, respectively. The developed RL-LAMP assay generally demonstrated a high sensitivity and short reaction time by detecting two-digit copy numbers of virus particles. Similar results were also reported on SARS-CoV-2 using RT-LAMP assay [14,28,29].

### 3.3. Testing the RT-LAMP Assay against the Clinical RNA Samples

The previous test aimed to determine the most sensitive target primer candidate in both genes (*ORF1a* and *N*). The primer targeting the B region in both genes demonstrated the highest sensitivity for the DNA and RNA. Ninety-six clinical samples (i.e., RNA) were tested to validate this RT-LAMP assay on actual clinical samples. All samples were positive for SARS-CoV-2 using RT-PCR assay with quantification cycle (Cq) values ranging from 14 to 34, as presented in Table 1.

The RT-LAMP assay was performed with 3 μL of the RNA clinical samples and incubated for 30 min against *ORF1a* (Figure 8) and *N* (Figure 9) genes. The results are summarized in Table 2, which is divided into two groups depending on the Cq values: group 1, Cq value < 30, and group 2, Cq value ≥ 30. The RT-LAMP assay exhibited high sensitivity when the Cq values of the RT-PCR were <30, and it can detect almost 95% of the positive samples. However, RT-LAMP assay sensitivity dropped to detect around half of the positive samples when the virus load was low (Cq value ≥ 30).

The developed RT-LAMP assay was validated with 96 positive RNA clinical samples. Yet, more clinical samples are needed for further validation to assess the sensitivity and specificity of the assay, particularly with samples that are characterized by a low copy number of the viral RNA. Moreover, negative samples are also needed to calculate the assay specificity, which is essential to avoid false-positive results. Clinical validation with RT-LAMP demonstrated approximately 95% for both genes, and the sensitivity dropped when tested against samples with Cq > 30 values. This drop in sensitivity of the RT-LAMP was also reported in the literature. Numerous studies on RNA clinical samples with a high Cq value (>30), in which the results did not pick up by the RT-LAMP assay or took a longer time > 35 min to be detected, which could affect some negative samples and might cause false-positive results [32,33,34]. Other studies found the cutoff of RT-LAMP when the Cq value was 32 or more [35,36]. Therefore, further optimization is needed to identify the Cq breakpoint for this assay and to improve it by increasing the input amount of RNA in the reaction.

### 3.4. Optimizing the RNA Extraction from Clinical Samples to the RT-LAMP Assay (Point of Care Testing)

The RNA extraction in the clinic setting required kits and particular types of machinery, which were not compatible with the simplicity of the RT-LAMP assay. To optimize the RNA extraction, the RNAse P is used regularly in SARS-CoV-2 diagnosis as an internal control. To address that, different methods for RNA extraction using two variants, i.e., temperature and proteinase K, on two types of samples: buccal swab and saliva. First, the sample was added directly to the RT-LAMP reaction to confirm whether the temperature (65 °C in 30 min) could trigger the RNA’s release and amplify the target (RNAse P). However, this outcome was not achieved from saliva or buccal swab (Figure 10A3), which was also confirmed by gel electrophoresis (Figure 10B3). Next, the samples were heated at 55 °C for 15 min, then 98 °C for 5 min in the presence of proteinase K. The result showed the reaction achieved (Figure 10A7,B7). Adding Tris HCl buffer (pH 8.0) instead of PBS to the samples during the extraction step caused the inhibition of the RT-LAMP reaction (Figure 10A8,B8). A high concentration of Tris HCl buffer reported could have interacted and slowdown the amplification of RT-LAPM by inhibiting the *Bst* DNA polymerase activity [29,37]. Interestingly, heating the samples without proteinase K was efficient for the RNA release in the saliva sample rather than the buccal swab, as shown in Figure 10A5,B5 and Figure 10A6,B6, respectively.

The minimum time required for the RNA release with proteinase K and heat was furtherly assessed. A low temperature of 55 °C for 5 min with proteinase K yielded a negative reaction (i.e., pink color) for the buccal swab and a positive reaction (i.e., yellow color) for the saliva sample (Figure 11A1,B1). By testing different time and temperature points with the presence of proteinase K, it was observed that heating the samples (saliva and buccal swab) for 5 min at 98 °C resulted in a positive reaction for both samples (Figure 11A2,B2). Similarly, by heating the samples in two successive conditions; at 55 °C for 5 min, then 98 °C for 5 min (Figure 11A3,B3), 55 °C for 10 min, then 98 °C for 5 min (Figure 11A4,B4), and 55 °C for 15 min 98 °C for 5 min (Figure 11A5,B5) all produced positive reactions. In addition, the potential of directly using proteinase K for RT-LAMP reaction with different volumes, i.e., 1, 3, and 5 μL, was also evaluated as shown in Figure 11A6, Figure 11A7 and Figure 11A8, respectively. The results demonstrated a positive reaction for the saliva samples at all volumes, while the buccal swabs produced a positive reaction at 3 and 5 μL volumes. Despite this color change, the gel electrophoresis did not amplify, as shown in Figure 11B6, Figure 11B7 and Figure 11B8, respectively, indicating a false-positive reaction. This might be due to the pH alteration by proteinase K, which would require a thorough investigation.

RNA extraction is vital for assays based on detecting nucleic acids [38]. Hence, the need to find a simple extraction method without using unique technique, equipment, or skilled personnel is preferable, particularly at point-of-cares with poor healthcare services. In this study, two RNA extraction methods onto two types of samples (saliva and buccal swab) were tested with RNAse P as an experimental control. The heat treatment at 98 °C for 5 min was efficient for releasing the RNA with the saliva sample before the RT-LAMP assay but less efficient with the buccal swab. Consequently, proteinase K was used with the heating treatment, which resulted in a positive reaction (yellow color) of RNA release in only 5 min. Similar results have been reported by Lalli et al. on saliva samples at 55 °C for 15 min, then 98 °C for 3 min with and without proteinase K. This study also reported that by spiking the virus particles on saliva, the method of extraction was able to detect 10^2^ viral copies by LAMP assay [39]. Another study on the Zika virus tested different RNA extraction methods before using the RT-LAMP assay. The finding indicated that treating the sample at 65 °C for 10 min and then 95 °C for 2 min with proteinase K gave a similarly positive result to the commercial RNA kit [40]. The commercialized lysis buffer already met the claim of the sample to RT-LAMP assay. Nevertheless, further investigation is needed to perform the extraction without using a manufactured kit, such as heat with proteinase K or RNA lysis buffer [39,41,42].

## 4. Conclusions

With the continuation of high COVID-19 transmission worldwide, there is still a need to constantly develop preventive or therapeutic tools and approaches to eradicate this pandemic. With the difficulty of delivering vaccines to countries with low incomes and poor healthcare services, there is still a gap in overcoming such diseases. Therefore, developing a diagnostic tool for SARS-CoV-2 with numerous advantages compared to other available virus detection assays, including its simplicity, rapidity, cost-effectiveness, sensitivity, and specificity, is needed. The developed RT-LAMP assay in this study has great potential to fulfill those criteria. The developed RL-LAMP assay showed high sensitivity in a short reaction time. The initial evaluation of five targets in *ORF1a* and *N* genes (A, B, C, D, and E). Unexplored regions in the *ORF1a* and *N* genes with novel primer sites have been identified. The primer design is a crucial step for nucleic acid amplification. Employing loop primers increased the assay specificity and reduced the reaction time. Furthermore, not all targets in *ORF1a* and *N* genes gave the same sensitivity, which may be due to the nature of the target gene or primer design factors. This assay detected a two-digit copy number of the viral RNA expressing its high sensitivity level. The developed RT-LAMP assay was validated against 96 positive RNA clinical samples, yet various clinical validations to ensure that this assay operates under different testing conditions are needed. Clinical validation with RT-LAMP demonstrated approximately 95% for both genes (*ORF1a* and *N*), and the sensitivity dropped when tested against samples with Cq > 30 values, which would require the assay for further optimization. However, this drop in sensitivity was also reported in previous studies. Currently, the sample to RT-LAMP assay was experimented with and validated. There is still an attempt to use a more straightforward RNA extraction method without additional techniques or equipment to ease the process of COVID-19 and other respiratory-based viral infection detection. Therefore, two RNA extraction methods for two types of samples (saliva and buccal swab) were investigated. It showed that using a heat treatment at 98 °C with proteinase K can result in a positive reaction (yellow color) of RNA release in 5 min only for both sample types. This rapid single-tube diagnostic tool was able to detect SARS-CoV-2. It can potentially be used as a high-throughput national screening for different respiratory-based infections within the Middle East region, such as the MERS virus or Brucella bacterium, particularly in remote and rural areas where lab equipment is limited. However, extensive clinical validation for the developed RT-LAMP is required as an essential step in the potential commercialization of this assay.

## Figures and Tables

**Figure 1 diagnostics-12-02232-f001:**
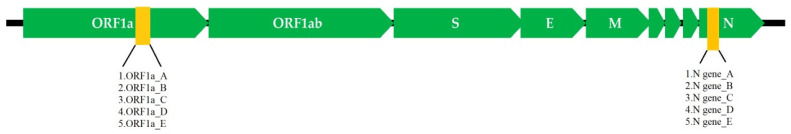
Illustration of SARS-CoV-2 genes. The diagram shows all genes presented in the virus genome. The *ORF1a* and *N* genes were targeted by the developed RT-LAMP assay, with five primers against each gene (A, B, C, D, and E) designed for that purpose.

**Figure 2 diagnostics-12-02232-f002:**
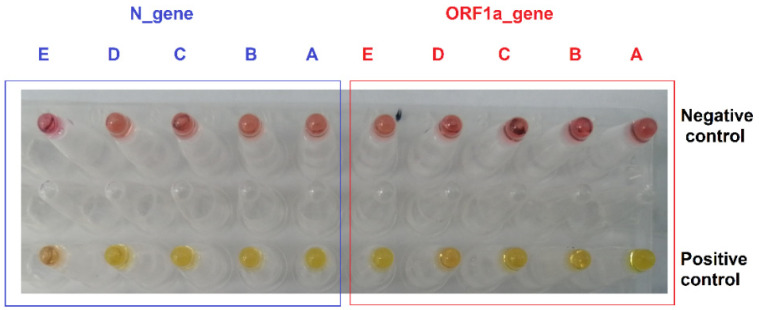
RT-LAMP with five primers targeting *ORF1a* and *N* gene using DNA fragment. The first row is the negative control reactions that contained no DNA template. In contrast, the second row is the RT-LAMP positive control assay performed against a synthesized viral cDNA fragment. The pink color indicates negative reactions, while the yellow denotes positive reactions.

**Figure 3 diagnostics-12-02232-f003:**
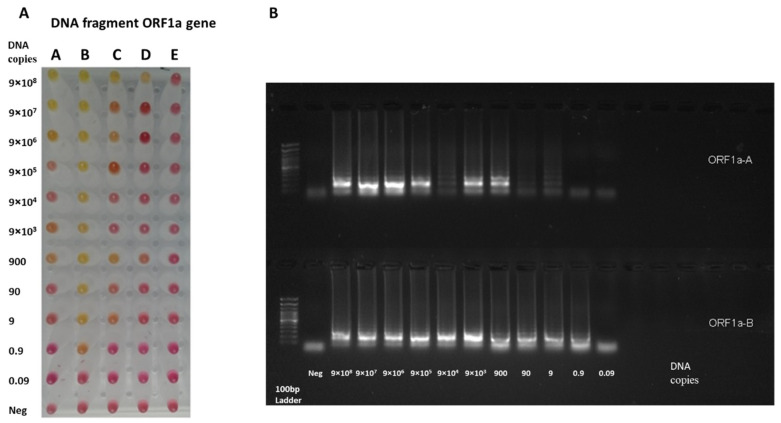
RT-LAMP sensitivity assay on *ORF1a* target using DNA samples. (**A**) The RT-LAMP assay was performed on DNA synthesized fragments. The assay was performed on five primer targets on the *ORF1a* gene (A, B, C, D, and E). The reaction was performed with a serial dilution of the *ORF1a* fragment, starting from 9 × 10^8^ to 0.09 copies. The reaction was incubated in the heat block at 65 °C for 30 min. (**B**) The gel electrophoresis for A and B targets the *ORF1a* gene. The B primer set has demonstrated a high level of sensitivity to detect one copy of the DNA fragment of the *ORF1a* gene.

**Figure 4 diagnostics-12-02232-f004:**
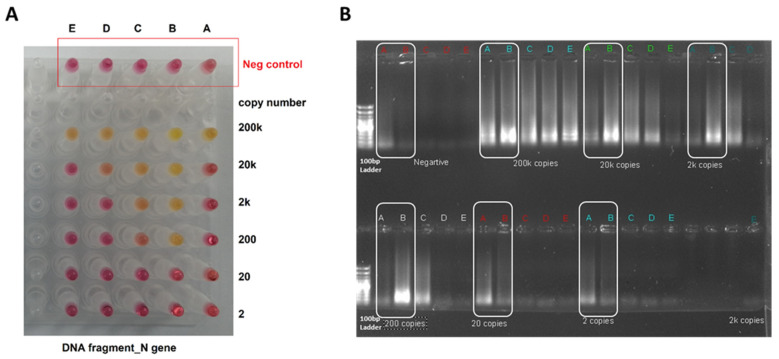
RT-LAMP sensitivity assay on *N* target using DNA samples. (**A**) The RT-LAMP assay was performed on DNA synthesized fragments. The assay was performed on five primer targets on the *N* gene (A, B, C, D, and E). The reaction was performed with serial dilution of the *N* fragment, starting from 200,000 to 2 copies. The reaction was incubated in a heat block at 65 °C for 30 min. (**B**) The gel electrophoresis for the A and B primer targets the *N* gene. The B primer set has demonstrated a high level of sensitivity to detect 200 copies of the DNA fragment of the *N* gene.

**Figure 5 diagnostics-12-02232-f005:**
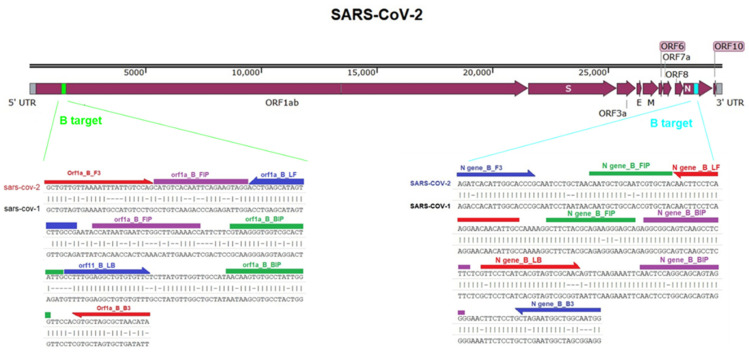
Comparison of the B target in the *ORF1a* and *N* gene of SARS-CoV-2 with similar regions in the SARS-CoV-1 genome. The Figure illustrates the location of the B target in the *ORF1* and *N* gene of SARS-CoV-2, which blasted against the SARS-CoV-1 genome sequence. The primers that were used in the RT-LAMP assay were denoted.

**Figure 6 diagnostics-12-02232-f006:**
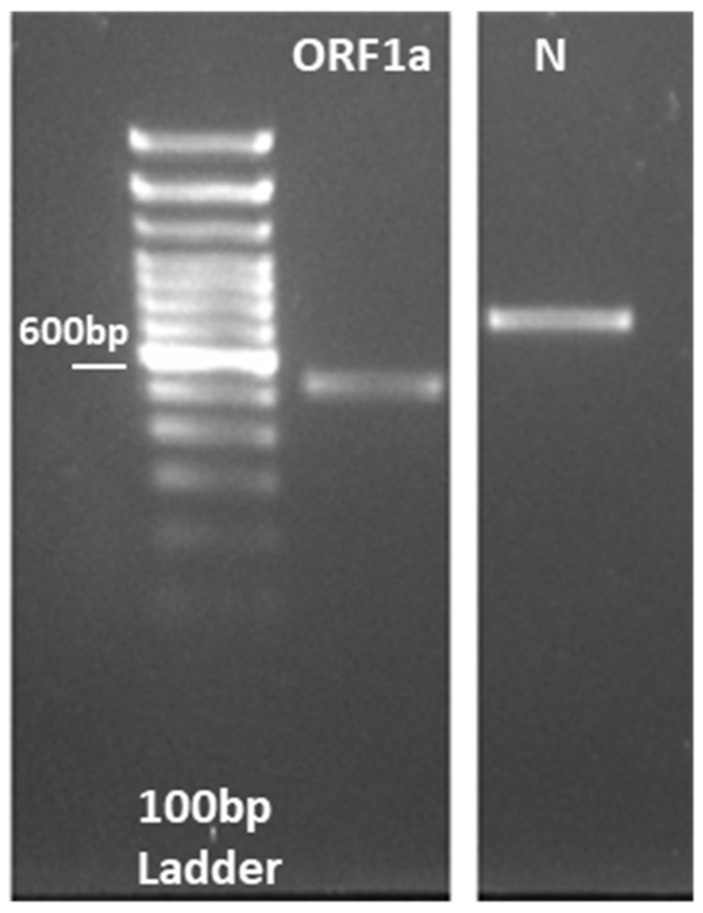
RNA synthesis by in vitro transcription, showing the gel electrophoresis for the PCR performed on ORF1a and N fragments where the T7 promoter sequence was added to the DNA fragment.

**Figure 7 diagnostics-12-02232-f007:**
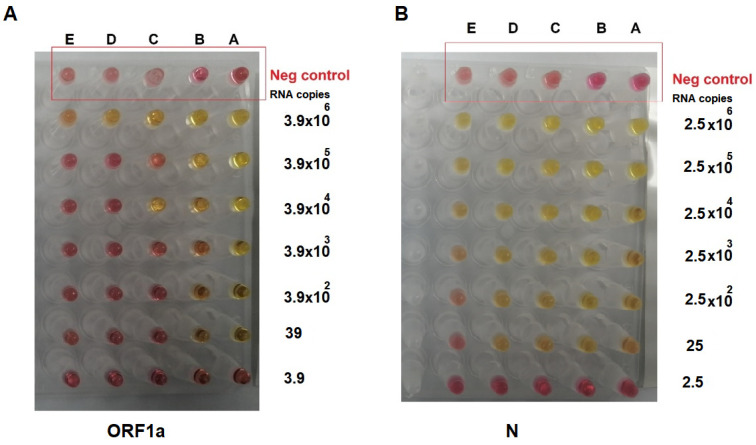
RT-LAMP assay on synthesized RNA molecules, showing (**A**) the RT-LAMP assay performed on synthesized RNA from the *ORF1a* gene and (**B**) the RT-LAMP assay performed on synthesized RNA from the *N* gene. The RNA molecules were serially diluted tenfold, and the reaction was incubated in the heat block at 65 °C for 30 min. The pink color indicated a negative reaction, while the yellow indicated a positive one. The RT-LAMP assay was highly sensitive as it detected 39 copies for *ORF1a* (primers A and B) and 25 copies for the *N* gene (primers A–D).

**Figure 8 diagnostics-12-02232-f008:**
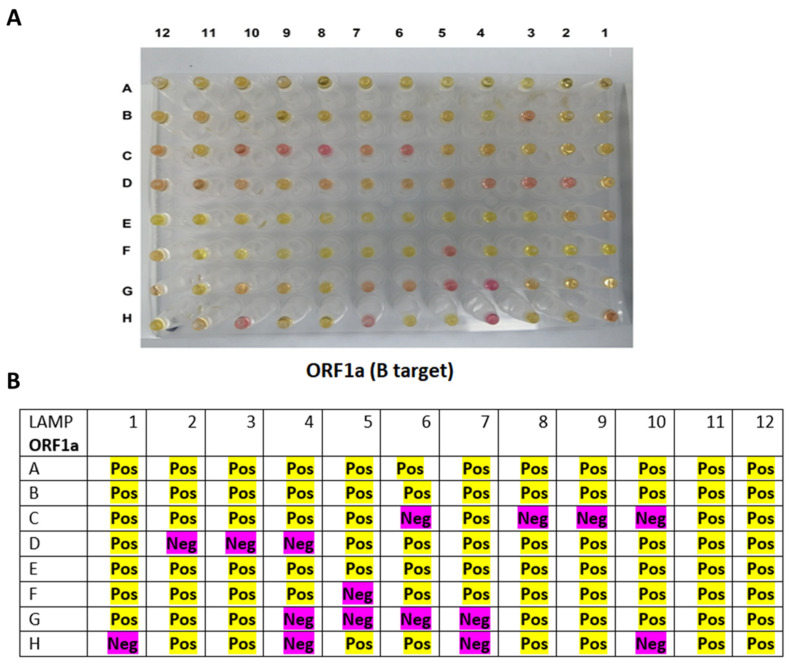
RT-LAMP assay on clinical RNA samples using primer targeting B region on the *ORF1a* gene. (**A**) The reaction results after incubation in a heat block at 65 °C for 30 min, in which the pink color indicated a negative reaction, while the yellow color denoted a positive reaction. (**B**) A table summarizing the RT-LAMP assay on the B target of the *ORF1a* gene. All of the positive samples with Cq values of <30 (n = 73) were detectable by 94.5% for the *ORF1a* gene using the RT-LAMP. The samples with Cq values of >30 were less sensitive to be detected by the RT-LAMP (47.8%).

**Figure 9 diagnostics-12-02232-f009:**
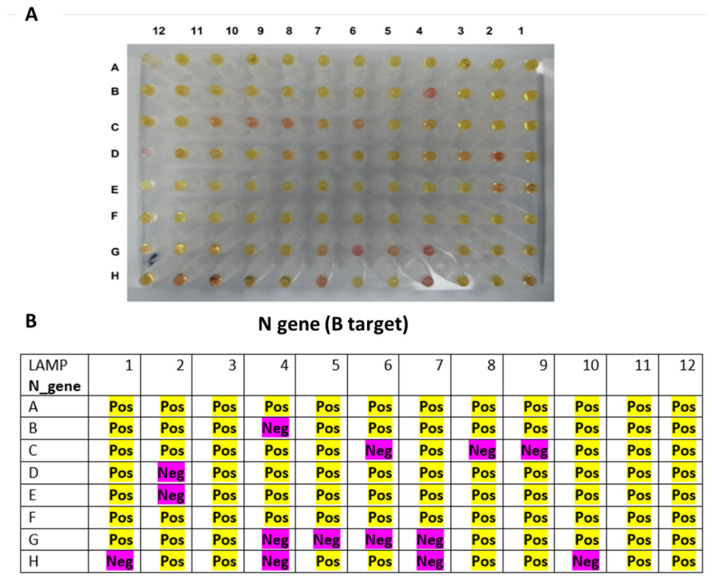
RT-LAMP assay on clinical RNA samples using primer targeting B region on the *N* gene. (**A**) The reaction results after incubation in a heat block at 65 °C for 30 min, in which the pink color indicated a negative reaction, while the yellow color denoted a positive reaction. (**B**) A table summarizing the RT-LAMP assay on the B target of the *N* gene. All positive samples with Cq values of <30 (n = 73) were detectable by 95.9% for the *N* gene using the RT-LAMP. The samples with Cq values of >30 were less sensitive to be detected by the RT-LAMP (52.2%).

**Figure 10 diagnostics-12-02232-f010:**
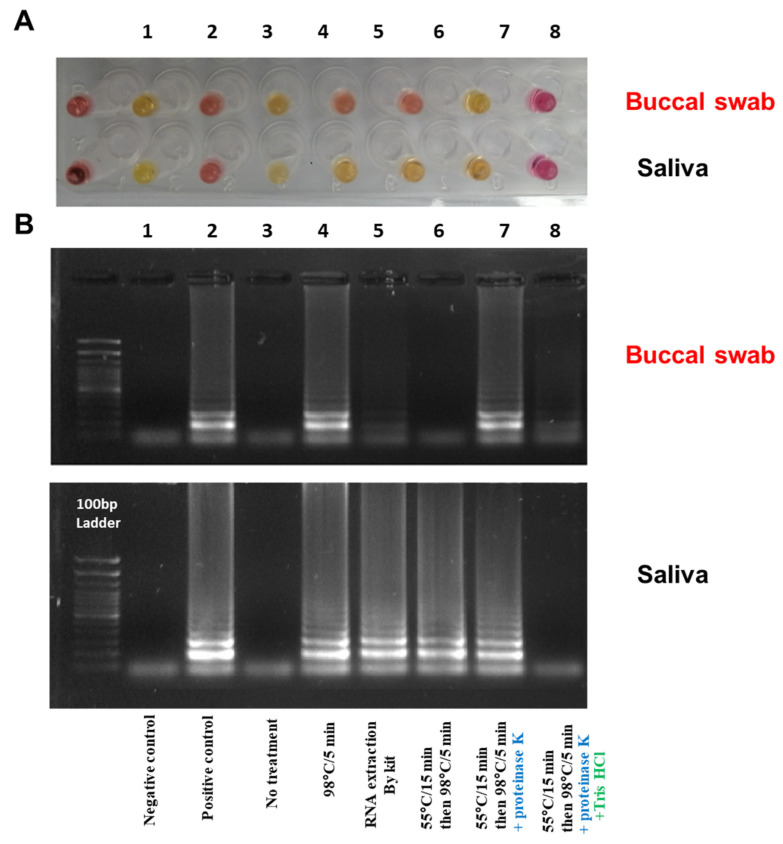
Optimizing RT-LAMP assay with different RNA extraction methods; (**A**) Using different extraction methods to show RT-LAMP on the RNAse P from the buccal swab and saliva sample; (**B**) Gel electrophoresis for the RT-LAMP assay resulted from A. Tube number (**1**) negative control (water), (**2**) positive control (RNA from COVID-19 patient), (**3**) sample without treatment, (**4**) RNA extraction by kit method (Monarch Total RNA miniprep Kit from NEB), (**5**) heat treatment for the sample at 98 °C for 5 min, (**6**) heat treatment for the sample at 55 °C for 15 min then 98 °C for 5 min, (**7**) heat treatment for the sample at 55 °C for 15 min then 98 °C for 5 min with proteinase K, (**8**) heat treatment for the sample at 55 °C for 15 min then 98 °C for 5 min with proteinase K in Tris HCl buffer. The reaction was incubated in a heat block at 65 °C for 30 min. The pink color indicated a negative reaction, while the yellow indicated a positive one.

**Figure 11 diagnostics-12-02232-f011:**
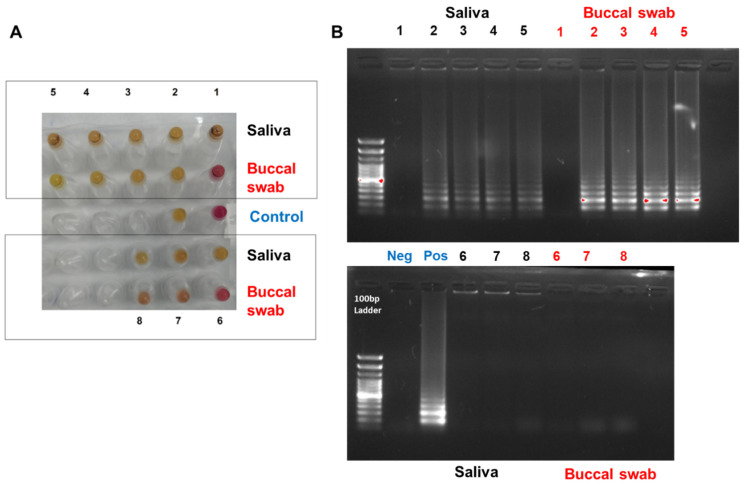
Optimizing RT-LAMP assay with proteinase K extraction method. (**A**) Show RT-LAMP on the RNAse P from buccal swab and saliva sample using proteinase K at different temperatures. (**B**) Gel electrophoresis for the RT-LAMP assay resulted from A. Tube number (**1**) heat treatment for the sample at 55 °C for 5 min with proteinase K, (**2**) heat treatment for the sample at 98 °C for 5 min with proteinase K, (**3**) heat treatment for the sample at 55 °C for 5 min then 98 °C for 5 min with proteinase K, (**4**) heat treatment for the sample at 55 °C for 10 min then 98 °C for 5 min with proteinase K, (**5**) heat treatment for the sample at 55 °C for 15 min then 98 °C for 5 min with proteinase K, (**6**) the sample (saliva or buccal swab) added directly to the RT-LAMP assay with 1 μL of proteinase k, (**7**) the sample (saliva or buccal swab) added directly to the RT-LAMP assay with 3 μL of proteinase k, (**8**) the sample (saliva or buccal swab) added directly to the RT-LAMP assay with 5 μL of proteinase k. All reactions were incubated in a heat block at 65 °C for 30 min for the RT-LAMP assay. Water and RNA from COVID-19 patients were used as negative and positive controls, respectively. The pink color indicated a negative reaction, while the yellow indicated a positive one.

**Table 1 diagnostics-12-02232-t001:** Clinical RNA plate with Cq value from RT-PCR assay.

Cq Value	1	2	3	4	5	6	7	8	9	10	11	12
A	27	22	33	14	21	19	14	18	25	25	21	29
B	15	17	27	21	18	20	18	33	18	27	30	25
C	29	24	18	31	23	29	21	33	25	28	25	24.5
D	17	33	33	27	23	22	17	24	14	27	24	32
E	23	32	24	17	26	21	24	31	21	30	16	31
F	28	19	26	17	32	19	20	27	26	17	24	24
G	24	20	30	34	34	33	34	19	27	25	16	34
H	32	21	19	32	18	19	34	15	22	34	29	15

**Table 2 diagnostics-12-02232-t002:** Data analysis from RT-LAMP on clinical RNA samples. The RT-LAMP was performed on the primer target B region of ORF1a and *N* genes. The newly developed RT-LAMP assay obtained comparable sensitivity to the RT-PCR assay. All positive samples with Cq < 30 values (n = 73) were detectable by 94.5% for the ORF1a gene and 95.9% for the *N* gene. The samples with Cq values > 30 designated either inconclusive or negative were less sensitive to RT-LAMP detection.

Targeted Gene	Cq < 30 (n = 73)	Cq ≥ 30 (n = 23)
*ORF1a*	69 (94.5%)	11 (47.8%)
*N*	70 (95.9%)	12 (52.2%)

## Data Availability

Not applicable.

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
