# Peer review of "Development and Validation of Reverse Transcriptase Loop-Mediated Isothermal Amplification (RT-LAMP) as a Simple and Rapid Diagnostic Tool for SARS-CoV-2 Detection"

_diagnostics, 2022, doi:10.3390/diagnostics12092232_

Round 1
Reviewer 1 Report
This manuscript submitted to Diagnostics describes the use of LAMP technology to detect SARS-CoV-2 to overcome limitations of molecular testing. While the methods and results are adequately described, there is significant concern regarding the readability of the manuscript. There are numerous gammer and typographical errors, which will need to be corrected. The introduction and abstract will require revisions, with emphasis on citing important papers already published on LAMP and SARS-CoV-2 and why this particular submission is similar or different (a PUBMED search for “loop-mediated” and “COVID” yields over 300 citations). While not comprehensive, a list of required changes or clarifications is listed here.
11. Abstract: COVID-19 is not an infection, SARS-CoV-2 is.
22. Abstract: provide details (numbers) of the sensitivity, specificity, and LOD in the abstract, not just a mention of them.
33. Intro: There is no recommendation to use serology to identify COVID-positive individuals
44. Mat and Method: What is meant by “second most important gene”?
55. Typo in Figure 2 legend: “raw”
66. Figure 6: what is in the “unused” lane marked “OR-2”?
77. Figure 7: What is the precision at the LOD? LOD dilution should be repeatedly assayed and shown to be positive 95% of the time.
88. Results: Breakpoint CT is 30? This is flawed logic. There is no scientific basis for this.
99. Only 3ul of RNA was used in the assay. Will using more increase sensitivity?
110. There are some low CT negatives by LAMP. Are these variants that are not picked up due to primer mismatch?
111. What is the effect of new variants to primer binding? Provide a detailed analysis comparing circulating strains and likelihood of assay failure.
112. Was Fig 8 and 9 repeated? What is the reproducibility?
113. Fig 10B needs labels on the figure.
Author Response
We thank the reviewer for their thoughtful and helpful comments. We attached the responses to each point below. Their suggestions have undoubtedly improved the paper, for which we are very grateful. Red font was used in all answers and amendments.
Reviewer 2 Report
The reviewed manuscript is dedicated to development of the new RT-LAMP for detection of SARS-CoV-2. Such colorimetric test could greatly facilitate testing in rural areas where advanced laboratory equipment is not available. Comparing to standard RT-PCR, RT-LAMP allows to perform DNA testing without sophisticated tools with equivalent sensitivity and specify. In the manuscript, authors described colorimetric RT-LAMP for detection of SARS-CoV-2 genomic RNA, which could be beneficial for developing countries and remote areas. However, a few comments need to be made concerning several topics.
Major issues:
1. Colorimetric real-time LAMP is described in numerous papers published since the first half of 2020. In the present work, authors skipped the survey of these articles in the introduction section. Moreover, authors didn’t explain how the advantages and novelty of their colorimetric RT-LAMP comparing to multiple past studies.
2. There are several serious issues in the experimental design.
a. Mutations under LAMP primers in SARS-CoV-2 genome were not taken in account. Since LAMP uses 6 primers (2 of them are long) instead of PCR, mutations could affect greatly the clinical sensitivity of the RT-LAMP assay in testing of constantly emerging new SARS-CoV-2 variants.
b. Higher sensitivity of ORF1a gene LAMP D, E., and a great increase of sensitivity for N gene LAMP C, D, E, when using RNA template. All LAMP for N gene were more sensitive for RNA target than for DNA target. This observation needs to be explained.
c. No negative samples were used to assess the clinical specificity of the RT-LAMP assay, and false-positive results rate remained unknown.
d. Page 10 “96 clinical samples (i.e., RNA) were tested to validate this RT-LAMP assay on actual clinical samples.” – how was RNA purified from clinical samples, what were these clinical samples and how they were collected and stored? What PCR-assay was used to test SARS-CoV-2?
e. Page 13 “In order to optimize the RNA extraction, the RNAse P that uses regularly in SARS-CoV-2 diagnosis as an internal control was used.” – why was RNAse P chosen instead of LAMP for SARS-CoV-2 and how were LAMP primers obtained for it?
f. Page 13 “Adding Tris HCl buffer (pH 8.0) instead of PBS to the samples during the extraction step caused the inhibition of the RT-LAMP reaction (Figure 10 A8 and B8).” – Tris-HCl buffer could change the pH of reaction which is crucial for the detection using pH-dependent dyes. In turn, PBS could not prevent changing of reaction pH caused by sample components, especially, from saliva. These issues are well-known for colorimetric LAMP with pH-dependent dyes and need to be clarify.
Minor issues:
Page 2 “In late 2019, Wuhan in Hubi Province in China had reported an unknown fast-spreading viral outbreak” – Hubei province
Page 2 “have caused two serious pandemics, such as Severe Acute Respiratory Syndrome (SARS-CoV) and Middle” – MERS outbreak was not considered as a pandemic by WHO.
Page 3 “The two genes (ORF1a and N gene) were synthesized in vitro by DNA Technologies, Inc. (IDT). Positive controls for the nucleocapsid gene (N gene, 1260 bp) and partial of ORF1 (510 bp) were synthesized.”
Page 3 “The transcribed RNA was purified by the Monarch RNA Cleanup Kit (New England Biolabs, U.K.) and then quantified by the SpectraMax QuickDrop spectrophotometer (Molecular Devices, San Jose, CA, U.S.A.).” – was there DNAse treatment?
Page 4 “12.5 μL of WarmStart LAMP 2X master mix, 1 μL of the sample, and the reaction” – was only 1 μL of sample analyzed in all RT-LAMP reactions?
Page 5 “The sensitivity of ORF1a was assessed from 9×108 to 0.09 copies” – copies per reaction or per μL of the sample? Was carrier added in dilution of the template to prevent binding to tube walls?
All DNA ladders are without lengths marks
Page 10 “cycle threshold (C.T.) values” – Cq in accordance with MIQE guide for real-time PCR
Page 10 “The RT-LAMP assay was performed with 3 μL of the RNA clinical samples” – the LAMP volume was 25 μL with 10 μL possible for the samples, why were only 3 μL added in LAMP reactions?
Author Response

(The authors gave the same response as above.)

Round 2
Reviewer 1 Report
All questions still not addressed. Proofing for English is still required.
COVID-19 is not a a virus therefore there is no such thing as a COVID-19 genome. Please use SARS-CoV-2 instead.
Abstract and Introduction are improved
Figure 6: what is in the “unused” lane marked “OR-2”?
Results: Breakpoint CT is 30? This is flawed logic. There is no scientific basis for this.
There are some low CT negatives by LAMP. Are these variants that are not picked up due to primer mismatch?
Was Fig 8 and 9 repeated? What is the reproducibility?
Fig 10B needs labels on the figure.
Author Response
Thank you for your comments. We respond to each point below in the attached file. Thank you!

Reviewer 2 Report
The authors have edited the manuscript scrupulously and with a great care. All questions mentioned in the review are solved; all statements and changes are supported by the results and cited articles. Many thanks to authors for their efforts and attention to all comments from the review. The only minor proposal is to increase the legend's font in all gel-electrophoresis figures. In the current size, it can be hard to read. After that, the manuscript can be published in the present form.
Author Response
We thank the refree for their comment. The size of legend's font in all gel-electrophoresis figures has been increased.
Round 3
Reviewer 1 Report
Thank you for addressing my concerns